# Pediatric Training Crisis of Emergency Medicine Residency during the COVID-19 Pandemic

**DOI:** 10.3390/children9010032

**Published:** 2022-01-01

**Authors:** Yan-Bo Huang, Yu-Ru Lin, Shang-Kai Hung, Yu-Che Chang, Chip-Jin Ng, Shou-Yen Chen

**Affiliations:** 1Department of Emergency Medicine, Chang Gung Memorial Hospital Linkou, Chang Gung University, Taoyuan City 333, Taiwan; yanhusuo79619@gmail.com (Y.-B.H.); mm200411800@cgmh.org.tw (S.-K.H.); changyuche@gmail.com (Y.-C.C.); ngowl@ms3.hinet.net (C.-J.N.); 2Department of Emergency Medicine, College of Medicine, Chang Gung Memorial Hospital Keelung, Chang Gung University, Taoyuan City 333, Taiwan; b117100030@tmu.edu.tw; 3Division of Medical Education, Graduate Institute of Clinical Medical Sciences, College of Medicine, Chang Gung University, Taoyuan City 333, Taiwan

**Keywords:** pediatric emergency medicine, emergency resident, COVID-19, emergency department

## Abstract

Coronavirus disease 2019 (COVID-19) is an emerging viral disease that has caused a global pandemic. Among emergency department (ED) patients, pediatric patient volume mostly and continuously decreased during the pandemic period. Decreased pediatric patient volume in a prolonged period could results in inadequate pediatric training of Emergency Medicine (EM) residents. We collected data regarding pediatric patients who were first seen by EM resident physicians between 1 February 2019, and 31 January 2021, which was divided into pre-epidemic and epidemic periods by 1 February 2020. A significant reduction in pediatric patients per hour (PPH) of EM residents was noted in the epidemic period (from 1.55 to 0.81, *p* < 0.001). The average patient number was reduced significantly in the classification of infection (from 9.50 to 4.00, *p* < 0.001), respiratory system (from 84.00 to 22.00, *p* < 0.001), gastrointestinal system (from 52.00 to 34.00, *p* = 0.007), otolaryngology (from 4.00 to 2.00, *p* = 0.022). Among the diagnoses of infectious disease, the most obvious drop was noted in the diagnosis of influenza and enterovirus infection. Reduced pediatric patient volume affected clinical exposure to pediatric EM training of EM residency. Changes in the proportion of pediatric diseases presented in the ED may induce inadequate experience with common and specific pediatric diseases.

## 1. Introduction

Coronavirus disease 2019 (COVID-19) is an emerging viral disease that has spread rapidly and caused a global pandemic. The patient volume of the emergency department (ED) has reduced with the progression of the pandemic [1,2,3]. Among ED patients, pediatric patient volume mostly and continuously decreased during the pandemic period, which has been noted in many countries [4,5,6]. A similar situation has been noted in Taiwan, although the epidemic was not as severe as in most countries [1]. This reduction could be partly attributed to parents’ fear of being infected [5,6,7,8]. Anti-epidemic policies, including wearing masks, keeping social distancing, and closing schools, which reduced the spread of diseases that commonly infect children, were other possible reasons [6]. All these factors contributed to the sustained reduction of pediatric patient volume of the ED.

Previous studies have revealed the impact of the COVID-19 pandemic on residency training in various aspects [4,9,10,11,12]. In addition to psychological stress of getting infected, a decrease in clinical experience was the most common problem of residency training in many specialties [13,14]. For emergency medicine (EM) residency training, clinical experience was important to foster the competency of residents [15]. Clinical experience based on seeing a variety of patients with different chief complaints and diagnoses is essential for the development of comprehensive competency and multitask ability [16,17]. Continued reduction of ED patient volume during the COVID-19 pandemic may decrease the clinical exposure of EM residents, especially for pediatric EM training. There were originally fewer pediatric than adult ED patients in most countries, and the pandemic induced a more severe decrease in the pediatric patient volume than that of adult patients [4,18]. This could result in inadequate pediatric training and experience for EM residents.

Our aim of this study was to explore the effect of epidemics on pediatric training of EM residents, including reduced patient volume and changes in disease distribution. Our study may provide further suggestions for pediatric training in EM residency during the pandemic.

## 2. Material and Methods

### 2.1. Study Design

This was a retrospective study of EM residents’ training in pediatric emergency medicine comparing the pre-epidemic and epidemic periods in Taiwan. To evaluate the influence of the COVID-19 epidemic on pediatric EM training of EM residents, we separated the study period into pre-epidemic and epidemic periods. The first COVID-19 case in Taiwan was confirmed on 11 January 2020, and the epidemic outbreak began in February 2020. The epidemic period in our study was defined as spanning from 1 February 2020 to 31 January 2021, and the pre-epidemic period was defined from 1 February 2019 to 31 January 2020. The study was approved by our institutional review board (Chang Gung Medical Foundation Institutional Review Board, IRB no. 202101493B0, passed on 31 August 2021).

### 2.2. Study Setting and Population

The study was conducted at Chang Gung Memorial Hospital, Linkou, a university-affiliated tertiary medical center with a 3600-bed capacity. It is one of the most large-scale children’s hospitals in Taiwan, with an estimated 36,000 pediatric patient visits annually.

There are 70 EM faculty members of Taiwan Society of Emergency Medicine (TSEM) in our ED and conducts an EM training program for 7–10 resident physicians annually. Each EM resident has EM training rotation structured by month. The EM residency program in Taiwan contained 3.5 years of training, including a 2-month pediatric ED rotation (one of the largest children’s hospital in North Taiwan), a 2-month pediatric ward rotation, and 1-year mixed department rotation. The study focused on the period of pediatric ED rotation of EM residents. The study included data spanning two years, from 1 February 2019 to 31 January 2021. Our study enrolled pediatric patients who were first seen by EM resident physicians at their pediatric ED rotation during the study period. EM resident shift schedules and working hours were collected and calculated. We collected information from the electronic medical record system, and the following data were recorded: patient gender, patient age, first EM resident physician who saw the patient, main diagnosis code, triage level, triage vital signs, referral, disposition, and length of stay. The diagnosis was decided by the main diagnosis code when the patients were discharged or admitted. The diagnoses were classified according to the International Classification of Diseases, Tenth Revision (ICD-10). Considering that R509 fever accounted for a large proportion of the main diagnosis code when patients were discharged from the ED, it was classified as an isolated category in our study.

### 2.3. Outcome Measurements

Data from patients who were first seen by EM resident physicians during the epidemic period were compared with those during the pre-epidemic period. Our primary outcome was the average number of pediatric patients seen by EM residents per hour (patients per hour, PPH) and the frequency of different diseases encountered and diagnosed per month in their pediatric EM rotation during the pre-epidemic and epidemic periods. The demographic characteristics and diagnosis categories according to different organ systems were also analyzed and compared between the pre-epidemic and epidemic periods.

### 2.4. Statistical Analysis

Data analysis was performed by using SPSS software (version 24.0 for Windows; SPSS Inc., Chicago, IL, USA). Regarding descriptive statistics, categorical variables are presented as numbers and percentages. The collected data of the patients were compared using Student’s t test and Mann–Whitney U test for continuous variables and Pearson’s chi-square test for categorical variables. A *p* value of <0.05 was considered statistically significant.

## 3. Results

A total of 6539 pediatric patients managed by 24 EM residents were included during the 2-year study period. The characteristics of the patients were analyzed and compared between the pre-epidemic and epidemic periods (Table 1). There was no significant difference in triage level (*p* = 0.241) between the pre-epidemic and epidemic periods. The proportion of referrals (6.51% vs. 8.31%, *p* = 0.006) and the admission rate (20.09% vs. 22.76%, *p* < 0.001) increased significantly in the epidemic period. In contrast, length of stay (LOS) decreased in the epidemic period (from 200.54 to 166.18 min, *p* < 0.001). A significant reduction in the PPH of EM residents was noted in the epidemic period (from 1.55 to 0.81, *p* < 0.001). Figure 1 shows the changes in pediatric ED patient volume and PPH of EM residents. Both decreased in the epidemic period, and PPH varied with patient volume.

The average number of patients classified according to organ systems seen by EM residents in each rotation was analyzed (Table 2). The average patient number was reduced significantly in the classification of infection (from 9.50 to 4.00, *p* < 0.001), respiratory system (from 84.00 to 22.00, *p* < 0.001), gastrointestinal (GI) system (from 52.00 to 34.00, *p* = 0.007), otolaryngology (from 4.00 to 2.00, *p* = 0.022), and soft tissue disease (from 2.50 to 2.00, *p* = 0.048). There was also a significant decrease in the diagnosis of fever (from 41.00 to 17.00, *p* = 0.037).

The top 10 diagnoses of pediatric patients seen by EM residents in the pre-epidemic and epidemic periods were calculated (Table 3). Influenza and pneumonia, which were noted in the top 10 pre-epidemic diagnoses, fell out of the rankings in the epidemic period. There were 1351 (3.57%) and 755 (1.99%) patients diagnosed with influenza and pneumonia in the pre-epidemic period, and the patient number dropped to 21 (0.12%) and 159 (0.88%) patients in the epidemic period, respectively. The use of the influenza virus antigen rapid test also decreased from 7654 times to 805 times in the epidemic period. Figure 2 illustrates the variation in common infectious disease number before and after the epidemic, including patients seen by EM residents (Figure 2A) and total patients (Figure 2B). Although all diagnoses of infectious disease were reduced in the epidemic period, the most obvious drop was noted in the diagnosis of influenza and enterovirus infection.

The age group distribution was calculated and compared (Figure 3). The proportion of neonates/infants and toddlers was elevated in the epidemic period. In contrast, the proportion of preschool- and school-age children decreased in the epidemic period. The dataset supporting the conclusions of this article is included within the article and its additional file (Appendix A).

## 4. Discussion

The decrease in pediatric experiential learning for EM residents is well known globally. Our study reported a further analysis of the reduced pediatric EM learning opportunities that our EM residency training program have experienced during the epidemic. Persistent pediatric ED volume reduction and variation of diseases diagnosed at ED in the COVID-19 pandemic may be attributed to anti-epidemic policies [5,6,19,20]. The decreased willingness of parents to bring children to the hospital because of fear of becoming infected with COVID-19 could be another reason [21]. Our study explored the influence of this effect on pediatric training in EM residency training by analyzing pediatric patients seen by EM residents. Although the total pediatric ED patient volume and variation in diagnosis during the pandemic were studied in some research, they may not reflect the real clinical learning of residents [5,6,19]. Some studies revealed that residents were under psychological stress during the pandemic, including fear of getting infected or personal protective equipment shortages, which could decrease their willingness to see patients in the epidemic [13,14,22,23]. In addition, the anti-epidemic policies of hospitals may restrict patients with COVID-like symptoms in specific areas, which may reduce the opportunity of residents to see certain patients. Consequently, direct analysis of patients treated by EM residents during the epidemic was necessary to determine the real impact of the epidemic on pediatric training.

Compared to the pre-epidemic period, our study showed that the number of pediatric patients seen by EM residents decreased significantly during the epidemic, and this reduction persisted even when the epidemic slowed down. This reduction almost completely varied with the total pediatric patient volume, which indicated that reduced patient volume was the main cause of decreased case exposure of EM residents during the epidemic. Insufficient clinical exposure may hinder residents from obtaining clinical competency and experiencing core EM diagnoses [24]. This situation could be more severe in pediatric EM training, as pediatric patient volume is usually less than that of adults [18]. Although pediatric ED volume increased gradually with epidemic alleviation in some countries, it was still lower than the pre-epidemic volume [19]. Prolonged decreased clinical exposure could severely impact pediatric EM training.

The EM residency program in Taiwan contained 3.5 years of training, including a 2-month pediatric ED rotation, a 2-month pediatric ward rotation, and 1-year mixed department rotation. About the pediatric training program, in America, a four-year residency includes at least 6 to 7-month pediatric training course (2.5 to 3.5-month block of PED, 1-month block of PICU, 1-month block of pediatric plastic surgery, 1-week block of pediatric anesthesia, and participating quarterly in high-fidelity simulation) [25]; in Canada, a five-year residency includes 7-month course (5-month block of PED, 1-month block of PICU, and 1-month block of pediatric anesthesia) [25]; in Singapore, a four-year residency includes 5.5-month course (5-month block of PED and 0.5-month block f PICU) [26]; in Colombia, a three-year residency includes 1 to 2-month course (PED and general ward) [27]. Comparing to other countries, pediatric EM training time in Taiwan seems less than other countries, but the pediatric case exposure seems not much less than other countries. A previous research reported an average of 723 pediatric patients seen by EM resident during their four-year residency in a hospital in North America, which was one of the largest providers of emergency care with yearly volume of approximately 115,000 visits [25]. From above data, the difference of pediatric patient number seen by EM residents between these two training systems is not significant, especially in the pre-epidemic period. Nevertheless, Taiwan has the lowest fertility rate in the world and decreased child population [28]. The impact of the pandemic on pediatric ED volume may further reduced pediatric case exposure of EM residents in Taiwan if the pandemic persisted.

Some previous studies have reported that the severity of the patients increased during the pandemic due to delays in seeing a doctor or policies such as moving constraints or quarantine [5,29,30]. In our study, there was no significant difference in triage level and vital signs, but the proportion of patients admitted to ward and ICU was elevated in the epidemic period. One possible explanation is that it was easier to get a bed during the epidemic period. This may be possible for the higher proportion of ward admission during the epidemic but may not explain the elevated percentage of ICU admission. The pediatric department of our hospital is one of the largest children’s hospitals in North Taiwan and the pediatric ICU beds were usually available in our hospital, even in the pre-epidemic period. Besides, the proportion of triage level one in the epidemic period was also higher than the pre-epidemic period, although there was no significant difference in overall triage level. From above reasons, it may imply that the proportion of pediatric patients with severe illness, who needed to be admitted to ICU, was really higher during the epidemic period. Residents may have the opportunity to learn about treating seriously ill patients in the epidemic if the patient volume was not too low.

Another important topic for pediatric EM training during the pandemic was the change in ED presentations and diseases of pediatric patients [6,7]. According to the analysis of patients seen by EM residents, the patient volumes of fever, infection, otolaryngology, respiratory system, and gastrointestinal system were significantly decreased. Among the common illnesses of pediatric ED patients, most are infectious diseases. Anti-epidemic policies, including wearing masks, washing hands, environmental disinfection, maintaining social distance, and closure of schools, may eliminate the spread of infection and decrease the occurrence of these diseases [20]. Influenza and lobar pneumonia largely declined, and some common infectious diseases of children, such as enterovirus infection, croup, and otitis media, were less common in the epidemic period. Diminished infectious disease also caused a change in age distribution, and the proportion of preschool and school-age patients who commonly develop these diseases decreased during the epidemic. This large reduction severely affected the clinical exposure of EM residents to these common pediatric infectious diseases, especially some diseases unique to children, such as croup. Inadequate clinical experiences could make learners have difficulty diagnosing and treating patients once these diseases return in the future. Some research has proposed that a lack of immune stimulation due to anti-epidemic interventions could induce an “immunity debt” and may have bad consequences when the pandemic is under control [20,31]. The reduction of infectious contacts secondary to hygiene measures may lead to decreased immune training in children and possibly to a greater susceptibility to infections in the future. Adjusting pediatric EM training courses and applying assisted teaching methods to enhance and compensate for the insufficient clinical experiences of EM residents was necessary.

In response to reduced patient volume and disappearance of certain infectious diseases, adopting innovative learning methods to strengthen pediatric EM training of EM residency are essential and warranted. Some studies have proposed that several innovative assisted teaching methods could help continue training for other subspecialties during the epidemic [32]. A common alternative is online learning, which includes online classes, webinars or online platforms [33,34]. It could be easily utilized for viewing educational videos of common pediatric disease like bronchiolitis, croup and asthma which incorporate presenting features, pathology, diagnosis, and management [35]. Although online learning could not completely replace hands-on training and clinical exposure, it was indicated to be an effective tool with a high degree of learner satisfaction [36,37]. High-fidelity simulation is another option to enhance clinical competency [38,39,40]. High-fidelity simulation training in pediatric education could be used to teach the skills managing with critical or rare situation such as neonatal resuscitation after delivery, cardiac arrest, shock or pediatric trauma [41,42]. Although high-fidelity simulation was usually used to learn critical ill patients, it could be designed for learning specific pediatric diseases diminishing in the pandemic. Learning about specific pediatric diseases could be achieved through appropriately designed scenarios in the absence of sufficient clinical cases. Virtual reality (VR)/augmented reality (AR) is also widely used to assist teaching methods in medical education [43,44]. It is often used to train residents to perform high-resuscitation procedures such as airway intubation or sedation [45,46,47]. They provided lifelike scenes and interaction with virtual objects for learners to practice. Other alternative learning solutions, such as the flipped classroom model, online practice questions, teleconferencing in place of in-person lectures, involving residents in telemedicine clinics, and the facilitated use of videos, were previously feasible and used for residency training [33]. These innovative solutions utilizing technology may help residents compensate for training deficiencies in pediatric EM during the COVID-19 pandemic. For the residents who experienced reduced pediatric patient volume during the epidemic, participating these new learning modalities actively is important besides clinical case exposure.

Our study has some limitations. First, this study was a retrospective study and conducted at single tertiary medical center teaching hospital; thus, selection bias could exist, and the results may not be applicable to other regions or countries. The influence of the pandemic varied in different countries. Data from other areas are necessary to evaluate the impact on pediatric EM education in other countries. Second, holiday factors that may affect ED volume were not included in our study. However, our study included a 1-year period in both the pre-epidemic and epidemic periods, which reduced this influence to a minimum. Third, our study focused on EM residents and did not include pediatric residents, so the situation may be different for pediatric residents. EM residents have a shorter course of pediatric EM training and less clinical exposure to pediatric patients than pediatric residents, and thus the impact of the pandemic would be larger for EM residents. Finally, an assessment of learning outcomes was not performed in our study, so the real effect of decreased clinical exposure is still unknown. Further research is needed to assess and compare the clinical performance of residents trained in the pre-epidemic and epidemic periods.

## 5. Conclusions

Reduced pediatric patient volume was significant during the pandemic and affected clinical exposure to pediatric training for EM residency. Changes in the proportion of pediatric diseases presented in the ED could induce inadequate experience of EM residents for common and specific pediatric diseases. Adopting assisted teaching methods by using innovative technologies to enhance pediatric EM training is necessary and could be a solution for the dilemma during the COVID-19 pandemic. Although COVID-19 pandemic will recede, these additional learning modalities could be integrated into training to achieve comprehensive learning outcomes.

## Figures and Tables

**Figure 1 children-09-00032-f001:**
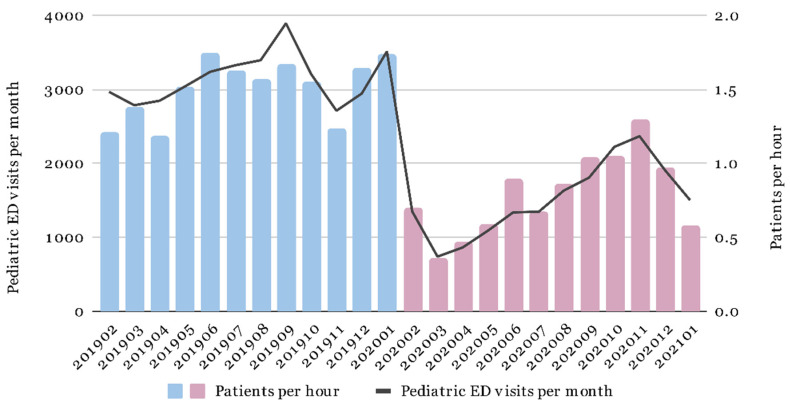
Pediatric patient volume and patients per hour (PPH) seen by EM residents in the pre-epidemic and epidemic periods.

**Figure 2 children-09-00032-f002:**
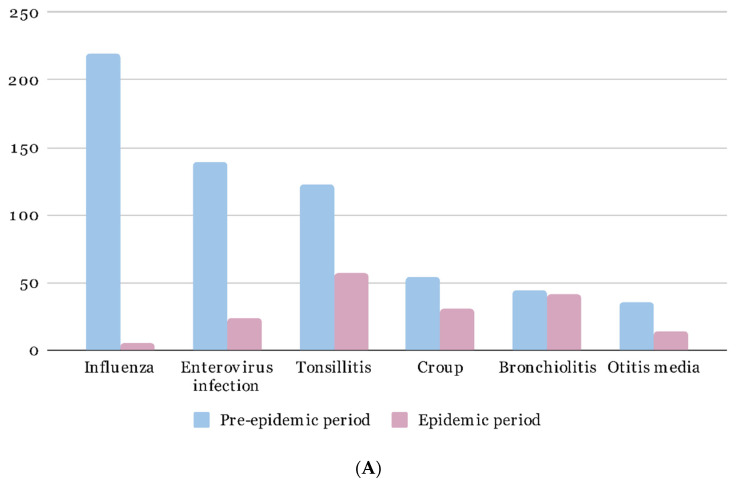
Number of patients with common pediatric infectious diseases in the pre-epidemic and epidemic periods. (**A**): Number of patients diagnosed by EM residents; (**B**): Total patient number.

**Figure 3 children-09-00032-f003:**
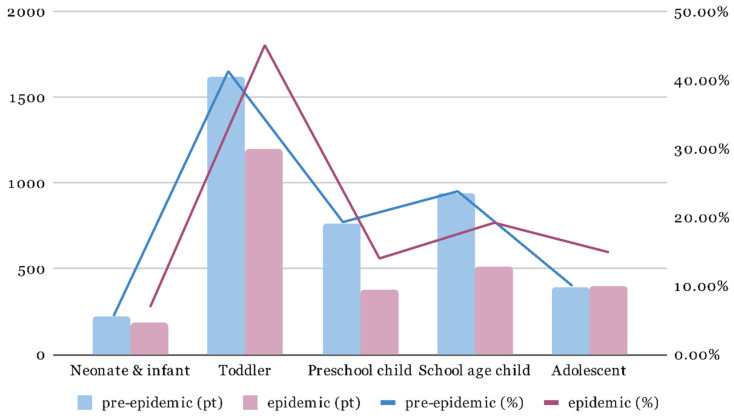
Age distribution of patients seen by EM residents. (Neonate & infant, 0 year; toddler, 1–3 year; preschool child, 4–5 year; school age child, 6–12 year; adolescent, 13–18 year).

**Table 1 children-09-00032-t001:** Characteristics of the pediatric patients seen by EM residents in the pre-epidemic and epidemic periods.

	Pre-Epidemic (*n* = 3903)	Epidemic (*n* = 2636)	
Age, mean ± SD (years)	5.11 ± 4.45	5.41 ± 5.17	*p* = 0.017 *
Sex (male)	2149 (55.06%)	1420 (53.87%)	*p* = 0.343
Triage level			*p* = 0.241
Level 1	184 (4.71%)	154 (5.84%)	
Level 2	1080 (27.67%)	702 (26.63%)	
Level 3	2408 (61.70%)	1639 (62.18%)	
Level 4	222 (5.69%)	135 (5.12%)	
Level 5	9 (0.23%)	6 (0.23%)	
Triage vital signs			
BT, mean ± SD (°C)	37.31 ± 1.23	37.27 ± 1.13	*p* = 0.214
HR, mean ± SD (beats per minute)	134.25 ± 30.86	130.52 ± 32.83	*p* < 0.001 *
RR, mean ± SD (breaths per minute)	23.7 ± 3.40	23.5 ± 4.51	*p* = 0.066
SpO_2_, mean ± SD (%)	96.16 ± 3.15	96.21 ± 1.96	*p* = 0.400
Referral	254 (6.51%)	219 (8.31%)	*p* = 0.006 *
Disposition			*p* < 0.001 *
Discharge	3032 (77.68%)	1936 (73.44%)	
AMA	43 (1.10%)	39 (1.48%)	
Admit to a ward	784 (20.09%)	600 (22.76%)	
Admit to ICU	26 (0.67%)	46 (1.75%)	
Transfer	7 (0.18%)	1 (0.04%)	
LOS, mean ± SD (minutes)	200.54 ± 260.20	166.18 ± 195.02	*p* < 0.001 *
PPH, median (IQR)	1.55 (1.22–1.64)	0.81 (0.57–0.96)	*p* < 0.001 *

ED, emergency department; BT, body temperature; HR, heart rate; RR, respiratory rate; SpO_2_, oxygen saturation; AMA, against medical advice; LOS, length of stay; PPH, patient per hour; *, indicates statistical significance, with p<0.05.

**Table 2 children-09-00032-t002:** Average patient number classified by organ systems in EM residents’ month rotation.

	Pre-Epidemic	Epidemic	
Infection	9.50 (6.00–20.50)	4.00 (3.00–5.00)	*p* < 0.001 *
Neoplasm	0.00 (0.00–1.00)	0.00 (0.00–1.00)	*p* = 0.612
Hematology	0.00 (0.00–1.00)	0.00 (0.00–1.00)	*p* = 0.683
Endocrine	0.50 (0.00–1.75)	0.00 (0.00–2.00)	*p* = 0.883
Psychology	1.00 (0.00–1.00)	1.00 (1.00–2.00)	*p* = 0.257
Neurology	6.00 (4.50–7.75)	5.00 (3.00–7.00)	*p* = 0.403
Ophthalmology	4.00(3.00–5.75)	3.00 (1.00–4.00)	*p* = 0.052
Otolaryngology	4.00 (3.00–7.00)	2.00 (1.00–4.00)	*p* = 0.022 *
Cardiovascular	2.00 (1.00–3.00)	3.00 (1.00–3.00)	*p* = 0.502
Respiratory	84.00 (53.00–111.00)	22.00 (17.00–28.00)	*p* < 0.001 *
Gastrointestinal	52.00 (39.50–57.00)	34.00 (20.00–45.00)	*p* = 0.007 *
Skin	11.00 (6.25–13.00)	8.00 (5.00–11.00)	*p* = 0.172
Soft tissue	2.50 (2.00–3.75)	2.00 (1.00–3.00)	*p* = 0.048 *
Urogenital	7.50 (6.00–12.00)	8.00 (4.00–11.00)	*p* = 0.635
Perinatal	0.00 (0.00–0.00)	0.00 (0.00–1.00)	*p* = 0.589
Congenital	0.00 (0.00–0.00)	0.00 (0.00–1.00)	*p* = 0.350
Toxin/Environment	6.00 (4.25–10.00)	7.00 (4.00–9.00)	*p* = 0.832
Other	3.50 (2.25–5.75)	3.00 (1.00–5.00)	*p* = 0.350
Fever	41.00 (24.00–53.73)	27.00 (19.00–36.00)	*p* = 0.037 *
Trauma	1.00 (0.00–1.00)	0.00 (0.00–1.00)	*p* = 0.333

*, indicates statistical significance, with *p* < 0.05.

**Table 3 children-09-00032-t003:** Top 10 diagnoses seen by EM residents in the pre-epidemic and epidemic periods.

No	Pre-Epidemic Top 10 Diagnosis		Epidemic Top 10 Diagnosis	
1	Fever	32.40%	Fever	37.70%
2	Noninfective gastroenteritis and colitis	12.70%	Noninfective gastroenteritis and colitis	19.80%
3	Influenza	9.90%	Acute upper respiratory infection	6.90%
4	Acute upper respiratory infection	9.50%	Bronchopneumonia	6.40%
5	Bronchopneumonia	8.30%	Nausea with vomiting	6.20%
6	Pneumonia	6.10%	Urinary tract infection	6.00%
7	Acute gastritis without bleeding	6.00%	Abdominal pain	5.30%
8	Acute tonsillitis	5.50%	Allergic urticaria	4.80%
9	Nausea with vomiting	5.20%	Acute tonsillitis	3.90%
10	Urinary tract infection	4.40%	Acute gastritis without bleeding	3.10%

The gray background color indicates the change in the top 10 diagnoses.

## Data Availability

The data in this study are available from the corresponding author upon reasonable request.

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
