# Peer review of "Pediatric Training Crisis of Emergency Medicine Residency during the COVID-19 Pandemic"

_children, 2022, doi:10.3390/children9010032_

Round 1

Reviewer 1 Report

Overall, I think this article is on an important topic to look at adequacy of paediatric exposure/experience during EM residency training in a changing times and the impact the Covid-19 pandemic has had on training. 

Page 2, line 58-60: Your study compensates for the knowledge gap - How??? you need to clarify what you mean here or reword this sentence

Page 2: Study setting and population - you describe a EM residency training scheme for 3 1/2 years of which only 2 months is in Pediatric ED (4.76%). This seems very little compared to other countries where longer periods are spent in Pediatric EM (e.g. UK and Ireland 6 months + as well as working in mixed departments on top of this during the training). Can you clarify what other paediatric experience / exposure trainees get during the remaining other time as it seems inadequate. This point also needs to be mentioned in the discussion to compare your countries training to other international systems in PEM exposure

Results: line 108-109 , in the period overall the average number of patients is only 272 per trainee which seems very low. Going back to my point above - this needs further clarification

Figure 1: no P value for admit to ICU. The difference between pre-epidemic and epidemic should also be commented on due to the nearly trebling in number during the Epidemic - was it easier to get a bed as this does not seem to correlate with triage acuity

Table 2: also adds further to adequacy of exposure with majority of trainees in both groups not seeing any patients with neoplasm, hematology or endocrine which is concerning. Please comment on this

Figure 2: the volume of patients with bronchiolitis pre-epidemic seems very low compared to influenza. Please comment on this as is it the bronch season came early, or is it just confirmed RSV positive cases and the data does not reflect a typical year or is this the normal pattern in Taiwan?

Discussions: line 159-161 is already said in the introduction and is repetitive. Discussion needs to commence with what your paper adds to the literature. 

The discussion needs to look at adequacy of Paediatric exposure and compare your system to other data from EM training in other countries as mentioned above. I would question if the trainees who see an average of 272 paediatric patients each have adequate PEM exposure as part of EM training so this needs to be clarified as no details on the exposure they get during the remaining of the training.

Author Response

Reviewer 1

Overall, I think this article is on an important topic to look at adequacy of paediatric exposure/experience during EM residency training in a changing times and the impact the Covid-19 pandemic has had on training.

Page 2, line 58-60: Your study compensates for the knowledge gap - How??? you need to clarify what you mean here or reword this sentence

ANS: Thank you for your comment. We have deleted this sentence to avoid misunderstanding.

Page 2: Study setting and population - you describe a EM residency training scheme for 3 1/2 years of which only 2 months is in Pediatric ED (4.76%). This seems very little compared to other countries where longer periods are spent in Pediatric EM (e.g. UK and Ireland 6 months + as well as working in mixed departments on top of this during the training). Can you clarify what other paediatric experience / exposure trainees get during the remaining other time as it seems inadequate. This point also needs to be mentioned in the discussion to compare your countries training to other international systems in PEM exposure

ANS: Thank you for your comment. We are sorry that we did not clarify the course. We have revised this part. Our EM residency training program included a 2-month pediatric ED rotation (one of the largest children’s hospital in North Taiwan), a 2-month pediatric ward rotation, and 1-year mixed department rotation. Our EM residents not only approached pediatric patients in this 2-month period of PED rotation. It also meets the criteria of Taiwan Society of Emergency Medicine. Please see Line 79-83.

Results: line 108-109 , in the period overall the average number of patients is only 272 per trainee which seems very low. Going back to my point above - this needs further clarification

ANS: Thank you for your comment. The average number of 272 is the number of 1 month PED rotation, including pre-epidemic and epidemic periods. Our EM residents had approached more than 800 pediatric patients during the 2-month PED course in the pre-epidemic period. We added the comparison between our course and other countries’. Please see Line 188-207.

Figure 1: no P value for admit to ICU. The difference between pre-epidemic and epidemic should also be commented on due to the nearly trebling in number during the Epidemic - was it easier to get a bed as this does not seem to correlate with triage acuity

ANS: Thank you for your comment.

We guess it means table 1 rather than figure 1.

Table 1 showed the P value for disposition, which was significant. In our hospital, pediatric ICU admission was usually available, even in the pre-epidemic period. Although there was no significant difference in overall triage level, the proportion of triage level 1 elevated in the epidemic period. Thus, the proportion of ICU admission was thought meaningful. For patients admitted to ward, it was possible that it was easier to get a bed during the epidemic period. We have revised our discussion to clarify this part. Please see Line 212-221. Thank you for your suggestion.

Table 2: also adds further to adequacy of exposure with majority of trainees in both groups not seeing any patients with neoplasm, hematology or endocrine which is concerning. Please comment on this

ANS: Thank you for your comment. The pediatric patients of neoplasm, hematology, and endocrine were relative rare at ED in Taiwan. For this reason, we have 2-month pediatric ward rotation to compensate the inadequate training of this part. The residents could approach these patients at wards. This may be different from other countries, and we hope you could understand this condition in Taiwan.

Figure 2: the volume of patients with bronchiolitis pre-epidemic seems very low compared to influenza. Please comment on this as is it the bronch season came early, or is it just confirmed RSV positive cases and the data does not reflect a typical year or is this the normal pattern in Taiwan?

ANS: Thank you for your comment. The diagnosis of bronchiolitis was made at pediatric ED and based on clinical symptoms. In contrast to Influenza, the RSV could not be detected at ED in Taiwan, so the number of diagnosis of bronchiolitis was usually less than influenza. The data reflect the real condition of Taiwan.

Discussions: line 159-161 is already said in the introduction and is repetitive. Discussion needs to commence with what your paper adds to the literature.

ANS: Thank you for your suggestion. We found the repetition and have shortened this sentence, but we kept the revised short sentence to continue the following paragraph. We hope you could understand. Please see Line 162-163. Thank you!

The discussion needs to look at adequacy of Paediatric exposure and compare your system to other data from EM training in other countries as mentioned above. I would question if the trainees who see an average of 272 paediatric patients each have adequate PEM exposure as part of EM training so this needs to be clarified as no details on the exposure they get during the remaining of the training.

ANS: Thank you for the comment. We added a new section discussing and explaining the issue in the Discussion. Please see Line 188-207. Thank you again for your valuable suggestion.

Reviewer 2 Report

My major concern about this paper is that it is not telling us anything that is not already known.  Every major medical center around the globe has had decreased Pediatric ED volumes.  Every EM residency is already well aware of the decreased Peds ED clinical encounters for their residents.  The suggestions that are made in this manuscript for enhancing clinical education are also already well known and the subject of many conversations.

Lines 58-59 you say: "Our study may compensate for the knowledge gap..."  That phrase should be deleted because you don't provide a means for this nor a demonstration of how it could be done.  You do go on to "provide further suggestions," so it is reasonable to leave that in the sentence.

Lines 190-193 represent probably misinterpretation and too much conjecture.  It is unlikely that patient severity increased, but rather, a greater proportion of lower acuity patients self-triaged to stay home.

Lines 224-225: These references are of questionable applicability since they are mainly about high-acuity patients, and the patient volume loss was mainly of low-acuity patients.

Author Response

Reviewer 2

My major concern about this paper is that it is not telling us anything that is not already known. Every major medical center around the globe has had decreased Pediatric ED volumes.  Every EM residency is already well aware of the decreased Peds ED clinical encounters for their residents.  The suggestions that are made in this manuscript for enhancing clinical education are also already well known and the subject of many conversations.

ANS: Thank you for your important comment. We understand that some articles have mentioned the condition of decreased PED volume and the change of diseases. However, there was no study focusing and analyzing on the data of residents. The consistency of patients seen by EM residents could be different from the proportion of total patient volume. For example, residents could escape to see high risk patients of COVID-19, and they may not approach some type of patients due to the anti-epidemic policy in some hospitals. We think direct analysis of data from residents is still valuable. We hope you could understand and consider the article. Thank you for your consideration.

Lines 58-59 you say: "Our study may compensate for the knowledge gap..."  That phrase should be deleted because you don't provide a means for this nor a demonstration of how it could be done.  You do go on to "provide further suggestions," so it is reasonable to leave that in the sentence.

ANS: Thank you for your correction. We have deleted the phrase to avoid misunderstanding.

Lines 190-193 represent probably misinterpretation and too much conjecture.  It is unlikely that patient severity increased, but rather, a greater proportion of lower acuity patients self-triaged to stay home.

ANS: Thank you for your comment. We have revised our manuscript to further clarify our point. Although the overall triage level is not significant different in the epidemic period, the proportion of acuity level 1 and patients admitted to ICU was higher. Please see Line 212-221.

Lines 224-225: These references are of questionable applicability since they are mainly about high-acuity patients, and the patient volume loss was mainly of low-acuity patients.

ANS: Thank you for your suggestion. We have revised the manuscript to clarify this condition. We added the notice for the condition: “Although high-fidelity simulation was usually used to learn critical ill patients, it could be designed for learning specific pediatric diseases diminishing in the pandemic”. Please see Line 254-256.

Round 2

Reviewer 1 Report

The authors have addressed the comments from my previous review in good detail and made the required updates.

There is a minor typo:

Line 204: Page 8 - "Nevertheless, Taiwan has the last total fertility rate in the world" - is this a typo and should read lowest rate? Consider rewording

I also note you use epidemic and pandemic on different parts of the paper - is it best to just use one term as Covid 19 is pandemic

Author Response

The authors have addressed the comments from my previous review in good detail and made the required updates.

There is a minor typo:

Line 204: Page 8 - "Nevertheless, Taiwan has the last total fertility rate in the world" - is this a typo and should read lowest rate? Consider rewording

ANS: Thank you for your correction. We have revised this part according to the suggestion. Please see Line 208.

I also note you use epidemic and pandemic on different parts of the paper - is it best to just use one term as Covid 19 is pandemic

ANS: Thank you for your comment. We used the word “pandemic” to indicate the global condition. In the other hand, we used the word “epidemic” to imply local condition in Taiwan or other countries. These two different terms were also used simultaneously in previous publications. We think the usage is appropriate, and we hope you could understand. Thank you for your suggestion again.

Reviewer 2 Report

You have to be clear that the decrease in Pediatric experiential learning for EM residents is already well known globally, and that your contribution is to provide a very specific breakdown of the reduced learning opportunities that your program experienced.

In terms of potential compensation for the loss of experiential learning, since you did not actually utilize any of these modalities, please put more effort into specifying which modalities could be used for which deficits match which modes (e.g. using video-based modules to address training about respiratory distress).

In your conclusion, address that the pandemic will recede, but that perhaps additional learning modalities can nevertheless be integrated into training.

Also, do you have any suggestions for the trainees who are now, or will soon, graduate not having had the Pediatric experience that they otherwise would have had?

Author Response

You have to be clear that the decrease in Pediatric experiential learning for EM residents is already well known globally, and that your contribution is to provide a very specific breakdown of the reduced learning opportunities that your program experienced.

ANS: Thank you for your comment. We have revised our manuscript and added this part in the beginning of our discussion. Please see Line 162-164.

In terms of potential compensation for the loss of experiential learning, since you did not actually utilize any of these modalities, please put more effort into specifying which modalities could be used for which deficits match which modes (e.g. using video-based modules to address training about respiratory distress).

ANS: Thank you for your valuable suggestion. We have expanded this part and revised our manuscript. Please see last paragraph, Line 249-276.

In your conclusion, address that the pandemic will recede, but that perhaps additional learning modalities can nevertheless be integrated into training.

ANS: Thank you for your suggestion. We have added this part into our conclusion. Please see Line 298-300.

Also, do you have any suggestions for the trainees who are now, or will soon, graduate not having had the Pediatric experience that they otherwise would have had?

ANS: Thank you for your suggestion. We suggest trainees participate the new learning modalities actively. We have added this into the discussion. Please see Line 274-276.